# Multidimensional analysis and detection of informative features in human brain white matter

**Adam Richie-Halford**[1]*, **Jason D. Yeatman**[2], **Noah Simon**[3], **Ariel Rokem**[1,4]

**1** eScience Institute, University of Washington, Seattle, Washington, United States of America, **2** Graduate School of Education and Division of Developmental and Behavioral Pediatrics, Stanford University, Stanford, California, United States of America, **3** Department of Biostatistics, University of Washington, Seattle, Washington, United States of America, **4** Department of Psychology, University of Washington, Seattle, Washington, United States of America

* richford@uw.edu

**Data Availability Statement:** The main software described in this study is available through GitHub at https://github.com/richford/AFQ-Insight. The version of software used in this study is also available at https://doi.org/10.5281/zenodo.4316000. To facilitate reproducibility and ease use

## Abstract

The white matter contains long-range connections between different brain regions and the organization of these connections holds important implications for brain function in health and disease. Tractometry uses diffusion-weighted magnetic resonance imaging (dMRI) to quantify tissue properties along the trajectories of these connections. Statistical inference from tractometry usually either averages these quantities along the length of each fiber bundle or computes regression models separately for each point along every one of the bundles. These approaches are limited in their sensitivity, in the former case, or in their statistical power, in the latter. We developed a method based on the sparse group lasso (SGL) that takes into account tissue properties along all of the bundles and selects informative features by enforcing both global and bundle-level sparsity. We demonstrate the performance of the method in two settings: i) in a classification setting, patients with amyotrophic lateral sclerosis (ALS) are accurately distinguished from matched controls. Furthermore, SGL identifies the corticospinal tract as important for this classification, correctly finding the parts of the white matter known to be affected by the disease. ii) In a regression setting, SGL accurately predicts "brain age." In this case, the weights are distributed throughout the white matter indicating that many different regions of the white matter change over the lifespan. Thus, SGL leverages the multivariate relationships between diffusion properties in multiple bundles to make accurate phenotypic predictions while simultaneously discovering the most relevant features of the white matter.

## Author summary

The connections between different parts of the brain form networks that are important for information transmission and for brain health. These connections are composed of nerve fibers that travel through the white matter portion of the brain. Thus, mapping tissue properties in white matter pathways can help us understand which connections and

of the software, the results presented in this paper are also provided in https://github.com/richford/afq-insight-paper as a series of Jupyter notebooks. We refer to four datasets in this work: ALS, WH, HBN, and Cam-CAN. The ALS dataset is openly available at https://doi.org/10.5281/zenodo.1161864. The WH dataset is openly available at https://doi.org/10.5281/zenodo.1161846. To facilitate acquisition of these data, we provide a 'make data' command in the paper repository at https://github.com/richford/afq-insight-paper. Data from the HBN study is available to access through: http://fcon_1000.projects.nitrc.org/indi/cmi_healthy_brain_network/. MRI data from this study is openly available. Phenotypic data can be freely accessed, upon establishing a data usage agreement with the Child Mind Institute, which is the data provider for this study. Data from the Cam-CAN study is available to access through: https://www.cam-can.org/index.php?content=dataset, and can be freely accessed upon agreeing to terms and conditions set forth by the Cambridge Centre for Ageing and Neuroscience, which is the data provider for this study.

**Funding:** ARH, AR, JDY and NS are supported by BRAIN grant 1RF1MH121868-01 from the National Institutes for Mental Health. ARH and AR are also supported by a grant from the Gordon & Betty Moore Foundation (https://www.moore.org/) and the Alfred P. Sloan Foundation (https://sloan.org/) to the University of Washington eScience Institute Data Science Environment. Computational resources for this study were provided by the Google Cloud Platform Academic Research Credits Program (AR). The funders had no role in study design, data collection and analysis, decision to publish, or preparation of the manuscript.

**Competing interests:** The authors have declared that no competing interests exist.

what tissue properties are relevant to brain diseases, or explain differences in behavior and cognition between different individuals. We developed a new statistical method that helps map the white matter by automatically identifying the features of the white matter that correspond to individual differences. Our approach relies on incorporating our knowledge of the anatomical sub-divisions of the white matter into the statistical model itself. We demonstrate that the model accurately captures differences between individual's with amyotrophic lateral sclerosis (ALS) and healthy controls. It also accurately captures brain changes that correspond with the ages of different individuals. We tested our method on four different datasets and found that it is more accurate than previous methods used with these datasets, while also allowing us to highlight which white matter tissue properties and which connections account for individual differences, thus providing interpretable results.

This is a *PLOS Computational Biology* Methods paper.

## Introduction

Non-invasive methods for measuring human brain structure and function have revolutionized our understanding of brain function. These measurements have demonstrated that interactions between networks of brain regions give rise to coordinated information processing and to the complex adaptive behavior that characterizes human cognition. Diffusion-weighted Magnetic Resonance Imaging (dMRI) provides a unique view into the physical properties of the connections that comprise these networks, by sensitizing the measurement to the directional diffusion of water in each voxel [1, 2]. Methods for computational tract-tracing from diffusion MRI, or tractography, combine the estimates of fiber orientations in each voxel to form streamlines that traverse the volume of the white matter [3, 4]. A variety of methods can be used to delineate the trajectory of major neural pathways among these streamlines [5]. *Tractometry* uses the results of tractography and models of tissue biophysics based on the patterns of diffusion in each measurement voxel to assess the physical properties of the white matter along specific pathways [6, 7]. In some previous tractometry-based studies, tissue properties along the length of each tract were summarized by taking the mean along each bundle, but there is a large body of evidence showing that there is systematic variability in the values of diffusion metrics along the trajectory of each bundle. This justifies retaining the individual samples along the length of each bundle [5, 6, 8, 9], hereafter referred to as tract profiling. While this retains important information about each individual's white matter, it also presents statistical challenges due to the dimensionality of the data. In past work, comparisons between groups or across individuals were done independently at each node of each bundle, for each diffusion metric. This approach is exhaustive, but statistical power is compromised by a multiple comparison problem [8, 10–12]. An alternative that circumvents the multiple comparison problem is to select just a few tracts to compare in each individual, or even segments of these tracts based on *a priori* hypotheses. This approach is appropriate when the biological basis of the process of interest is relatively well understood (for a recent example, see [13]). Sometimes, these approaches are combined: a bundle is selected based on *a priori* knowledge, and all the data in the bundle of interest are used together to fit a model that can predict differences between individuals [14].

The present work aims to balance predictive accuracy with descriptive power [15, 16] by capitalizing on all of the available data across all bundles, while also retaining and elucidating spatial information about the locations that are most informative for discriminative performance. Unlike many previous analyses of white matter (e.g., using TBSS [17] and connectometry [18]), which focus on classical inference about the differences between groups or individuals in terms of their white matter properties, the focus of the present work is on the combination of features that facilitate accurate prediction of the individual differences in a particular phenotype. For example, an accurate prediction of which individuals in a group have a particular disease, or a prediction of their age with small error. The distinct goals of *inference* and *prediction* are not necessarily incongruent, but can sometimes be in tension [19]. This distinction is also similar to the distinction between "encoding" and "decoding" used in the functional MRI literature [20]. In the present work, we predict the phenotypical variance in a group of subjects, or classify group membership, based on a linear combination of the features estimated with tract profiling.

Using this approach, we first need to deal with the large and asymmetric dimensionality of the data: tract profile data usually has many more features (i.e., number of measurements per individual) than samples (number of subjects), which makes inferences from the data about phenotypical differences between individuals ill-posed. This regime is the target of several statistical learning techniques, and is often solved by various forms of regularization.

The Lasso algorithm minimizes the sum of the absolute values of contributions of each feature [21]. This tends to shrink to zero the contributions of many of the features, providing results that are both accurate and interpretable. When additional structure is available in the organization of the data, regularization algorithms can take advantage of this structure. For example, if the features lend themselves to a natural division into different groups, the Group Lasso (GL) can be used to select groups of features, rather than individual features [22]. The Sparse Group Lasso (SGL) elaborates on this idea by providing control both of group sparsity, as well as overall sparsity of the solutions [23]. Because the features measured with tractomery lend themselves to grouping based on the tracts from which each measurement is taken, GL and SGL provide useful tools for linear model fitting in problems of this form. Here we develop an implementation of SGL that is well suited to the analysis of tract profile data. In addition, we demonstrate the power and flexibility of this approach by applying it to both classification and continuous prediction problems. Our data flow is represented in Fig 1 and explained in further detail in the Methods section.

One more approach to multivariate analysis of neuroimaging data is to perform inference or prediction in a transformed feature space rather then the original diffusion metric feature space. For example, in Lasso PCR, diffusion features can be projected onto a principal components (PC) basis for use in a sparsity constrained model [24, 25]. While these approaches achieve high predictive performance and may even yield biologically interpretable principal components [12], they neglect anatomical grouping information. We view these approaches as complementary to the SGL-based approach: PCR based approaches seek to model brain-phenotype relationships using the most parsimonious representation of variance in the diffusion measures. In contrast, our SGL-based approach seeks to establish whether prior knowledge of anatomical grouping improves modeling of brain-phenotype relationships. Motivated by this distinction, we also introduce a union of the two approaches called PCR-SGL, in which each group of features is independently transformed into its PC basis, thereby retaining the anatomical grouping information of the original feature space. We demonstrate circumstances in which this approach both helps and hinders predictive performance.

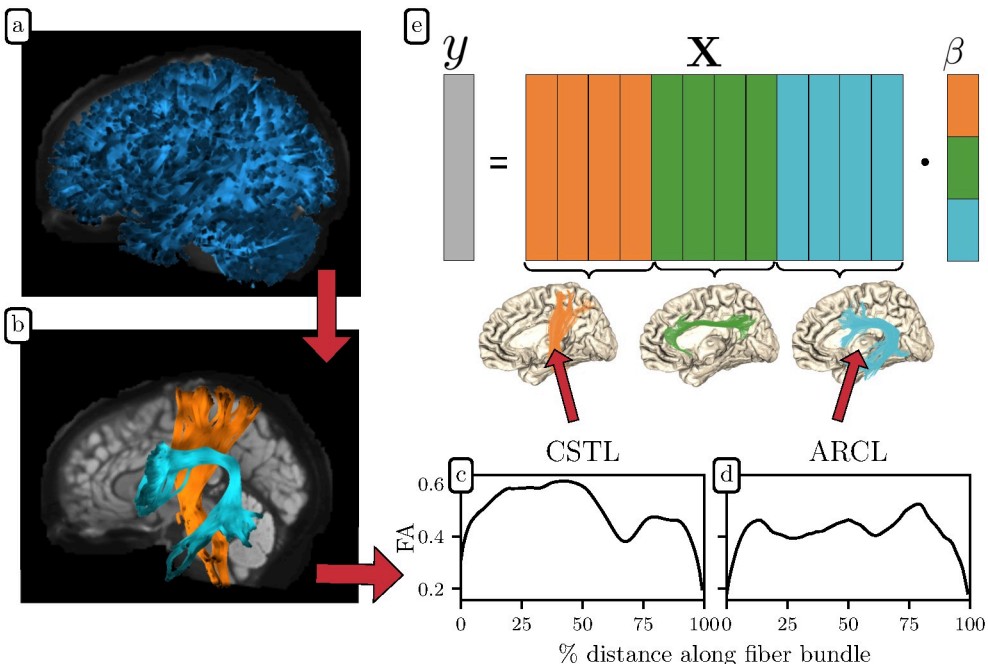

**Fig 1. Tractometry data flow. (a)** Whole brain tractography generates streamlines approximating the trajectories of white matter connections. **(b)** Tractometry classifies these streamlines into anatomical bundles. In this case, we show the left corticospinal tract (CSTL) and the left arcuate fasciculus (ARCL) over a mid-saggital anatomical slice. Tract profiling further extracts bundle profiles, quantifications of various diffusion metrics along the length of the fiber bundle. Here, we show one subject's fractional anisotropy (FA) profile for **(c)** the CSTL and **(d)** the ARCL. **(e)** the phenotypical target data and tract profile features can be organized into a linear model, $\hat{y} = \mathbf{X}\hat{\beta}$. The feature matrix $\mathbf{X}$ is color-coded to reveal a natural group structure: the left (orange) group contains $k$ features from the CSTL, the middle (green) group contains $k$ features from the left cingulum cingulate (CGCL), and the right (blue) group contains $k$ features from the ARCL. The coefficients in $\hat{\beta}$ follow the same natural grouping. Panels (a) and (b) are adapted from https://figshare.com/articles/figure/example_tractography-segmentation/14485350, and reproduced under the CC-BY license (https://creativecommons.org/licenses/by/4.0/).

## Results

We developed a method for analyzing dMRI tract profile data that uses the Sparse Group Lasso (SGL) to select features that are sparse both at the group (bundle) level, as well as overall. We demonstrate the use of this method on four different datasets in both a classification setting and a regression setting.

### SGL accurately detects ALS from tractometry data

Using data from a previous study of patients with amyotrophic lateral sclerosis (ALS) [26], we tested the performance of SGL in a classification setting. The previous study predicted ALS status with a mean accuracy of 80% using a random forest algorithm based on *a priori* selection of features only within the CST bundle-of-interest. SGL delivers improved predictive performance, with a cross-validated accuracy of 83% and an area under the receiver operating characteristic curve (ROC AUC) of 0.88, without the need for *a priori* feature engineering. We also predicted ALS diagnosis using the PCR-SGL, wherein each group of features is independently transformed to its PC basis and an SGL model is fit to the transformed features. This model achieves 88% accuracy and an ROC AUC of 0.9 (Fig 2a). In addition to this classification performance, both SGL and PCR-SGL also identify the white matter tracts most important for ALS classification. Fig 2b shows that PCR-SGL identified as potential disease biomarkers the

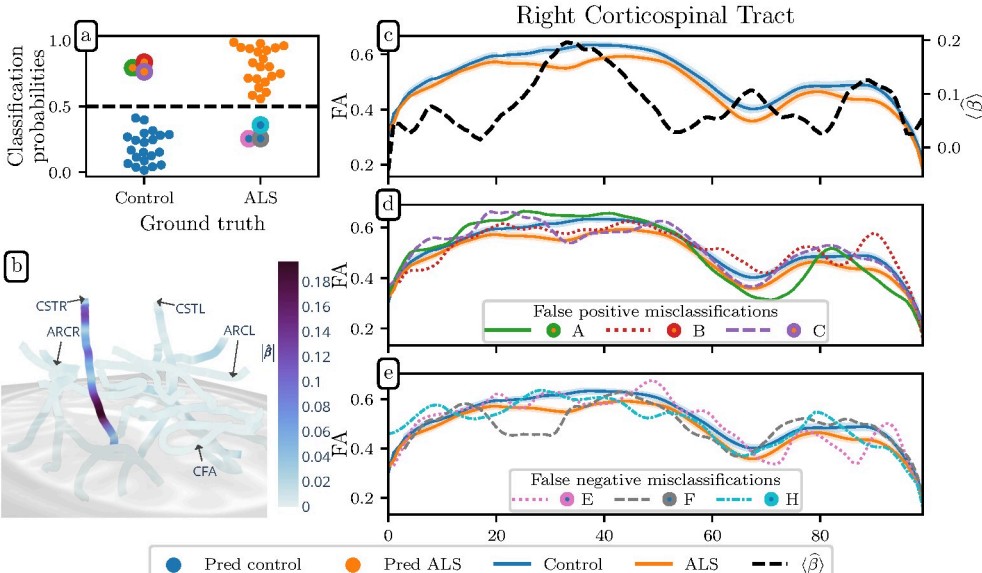

**Fig 2. PCR-SGL accurately and interpretably predicts ALS diagnosis. (a)** Classification probabilities for ALS diagnosis, with controls on the left, patients on the right, predicted controls in blue, and predicted patients in orange. That is, orange dots on the left represent false positives, while blue dots on the right represent false negatives. We achieve 83% accuracy with an ROC AUC of 0.88. **(b)** PCR-SGL coefficients are presented on the core fibers of major fiber bundles. They exhibit high group sparsity and are concentrated in the FA of the corticospinal tract (CST). The brain is oriented with the right hemisphere in the foreground and anterior to the right of the page. The CSTL, CSTR, callosum forceps anterior (CFA), left arcuate (ARCL), and right arcuate (ARCR) bundles are indicated for orientation. **(c)** PCR-SGL identifies three portions of the CST as important, where $\hat{\beta}$ (dashed line, right axis) has large values. These are centered around nodes 30, 65, and 90, corresponding to locations of substantial differences in FA between the ALS and control groups (shaded areas indicates standard error of the mean). **(d)** Bundle profiles for false positive classifications. Line colors correspond to the marker edge color in the top left plot. These individuals have reduced FA in the CST portions which SGL identified as important. Their misclassification is coherent with the feature importance and the group differences in FA. **(e)** Individual bundle profiles for false negative classifications. These individuals have bundle profiles which oscillate between the group means.

diffusion measures in the CST from the cerebral peduncle to the corona radiata, agreeing with the previous study from which these data were extracted [26]. The relative importance of white matter features is captured in the $\beta$ coefficients from Eq (3). Fig 2c depicts these coefficients along the right CST, plotted over the FA values for the control and ALS subject groups (see S1 and S2 Figs for tract profiles of all 18 tracts in these groups). We find that SGL and PCR-SGL select FA metrics in the corticospinal tract and particularly in the right corticospinal tract as most important to ALS classification, confirming previous findings [27–36] and identifying the portions of the brain that were selected *a priori* in the previous study from which we obtained the data [26].

To assess the added value of incorporating anatomical knowledge into our model, we compared our classification results to those obtained from four other models: a pure lasso model, an elastic net model, a bundle-mean lasso (i.e. a lasso model trained only on the mean metric values from each bundle), and a principal components regression lasso (Lasso PCR) model [24, 25]. These models achieved accuracies of 76%, 76%, 79.5%, and 71.5% respectively and ROC AUC values of 0.76, 0.75, 0.82, and 0.71, respectively using the same cross-validation strategy used to assess the SGL model. This difference in performance justifies the additional complexity of the SGL over simpler sparsity-inducing regression strategies. The relatively poor performance of Lasso PCR suggests that features with small variance are nonetheless relevant

for predicting ALS diagnosis. Or equivalently, that group differences between ALS diagnoses are not the dominant source of variance in the diffusion metrics.

The $\beta$ coefficients exhibit high bundle level sparsity; only some bundles are important, which can be be confirmed by observing the value of $\alpha$, the regularization hyperparameter that controls the mixture of the Group Lasso and lasso penalties, selected through nested cross-validation (see Eq (3)). If $\alpha$ is closer to zero, it indicates that the phenotype in question preferentially correlates with only a few groups of covariates. For the ALS dataset, the SGL model has $\alpha$ = 0.21 and the PCR-SGL model has $\alpha$ = 0.4, confirming that the white matter correlates of ALS reside mostly in one bundle, namely the CST.

Analyzing the ways in which the model mislabels individuals also provides insight. We found that mislabelled subjects are outliers relative to their group with respect to diffusion features of the CST (Fig 2d and 2e). The false positive classifications have reduced FA in one or more of the three sections of the CST where $\|\hat{\beta}\|$ is large in Fig 2c. The false negative subjects have FA profiles that oscillate between the two group means. Thus, when the SGL method predicts incorrectly this is done in a comprehensible manner.

## SGL accurately predicts age from tractometry data

To test the performance of SGL in a continuous regression task, we focus here on the prediction of biological age in three datasets named WH, HBN, and Cam-CAN (see the Methods section for a description of each dataset). Prediction of "brain age" is a commonly undertaken task in neuroimaging machine learning, in part because these predictions, and deviations therefrom, may be diagnostic of overall brain health (for a review, see Cole et al. [37]). However, as Nelson et al. [38] have observed, aging biomarkers are subject to unique challenges and tend to be noncausative. Our interest in aging here relies on its utility as a methodological benchmark. Biological age operates on a natural scale, with meaningful and easily understood units, and it's popularity as a machine learning target makes it valuable for comparisons with other studies.

The WH [39], HBN [40], and Cam-CAN [41, 42] datasets used here contain data from 76, 1651, and 640 subjects, respectively, ranging from 6–50, 5–21, and 18–88 years of age, respectively. In each case, biological age was used as the predicted variable $y$. SGL was fit to the tract profile features FA and mean diffusivity (MD) in 18 major brain tracts, with diffusion metrics extracted from diffusion tensor imaging (DTI) for the WH dataset and diffusion kurtosis imaging (DKI) [43] for the HBN and Cam-CAN datasets see S3–S8 Figs for the full tract profile information in all tracts/datasets.

To evaluate the fit of the model, we used a nested cross-validation procedure. In this procedure, batches of subjects are held out. For each batch (or fold), the model is fully fit without this data. Then, once the parameters are fixed, the model is applied to predict the ages of held out subjects based on the linear coeffiecients. This scheme automatically finds the right level of sparsity and fits the coefficients to the ill-posed linear model, while guarding against overfitting. SGL accurately predicts the age of the subjects in this procedure, with a median absolute error of 2.67, 1.45, and 6.02 years for the WH, HBN, and Cam-CAN datasets, respectively and coefficients of determination $R^2$ = 0.52, 0.57, and 0.77, respectively (see Fig 3, top panels). The predictions for Cam-CAN are competitive with a recent state of the art prediction [44], which used streamline density to estimate the brain's structural connectivity and achieved $R^2$ = 0.63. The median absolute errors are also lower than the results of a recent study that predicted age in a large sample that included the Cam-CAN data, and was based on diffusion MRI features [45].

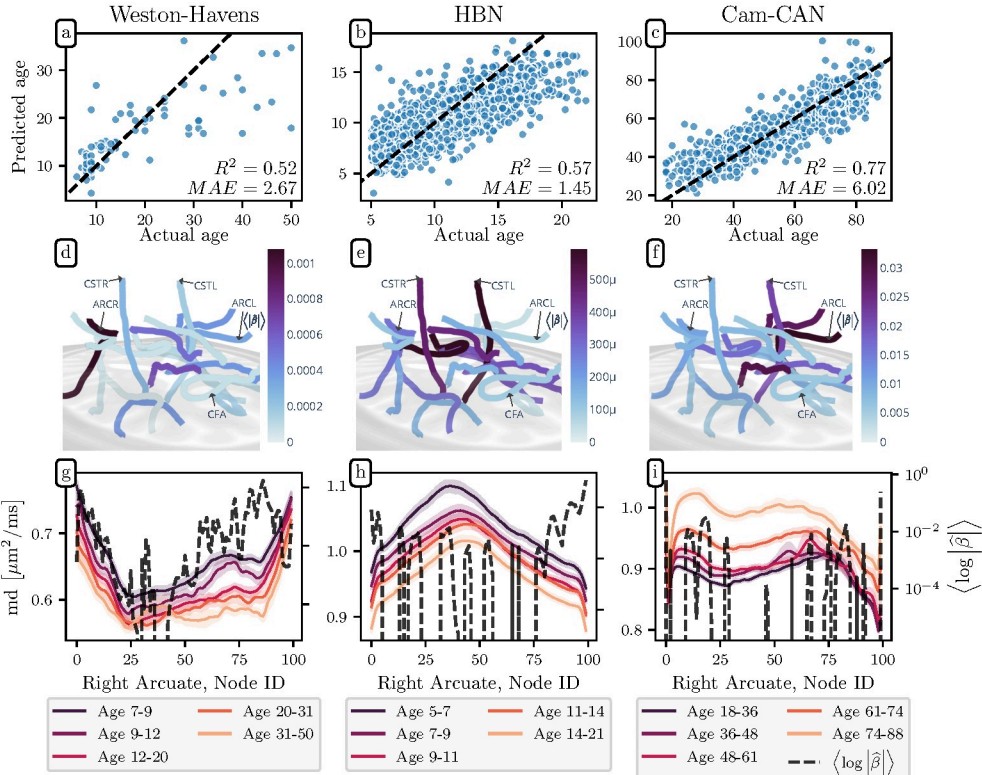

**Fig 3. Predicting age with tractometry and SGL. (top)** The predicted age vs. true age of each individual from the test splits (i.e., when each subject's data was held out in fitting the model) for the **(a)** WH, **(b)** HBN, and **(c)** Cam-CAN datasets; an accurate prediction falls close to the $y = x$ line (dashed). The mean absolute error (MAE) and coefficient of determination $R^2$ are presented in the lower right of each scatter plot. **(middle)** Feature importance for predicting age from tract profile in the **(d)** WH, **(e)** HBN, and **(f)** Cam-CAN datasets. The orientation of the brain is that same as in Fig 2b, however because the coefficients exhibit high global sparsity (as opposed to group sparsity), we plot the mean of the absolute value of $\hat{\beta}$ for each bundle on the core fiber. The global distribution of the $\hat{\beta}$ coefficients reflects the fact that aging is not confined to a single white matter bundle. **(bottom)** Age quintile bundle profiles for the **(g)** WH, **(h)** HBN, and **(i)** Cam-CAN datasets.

In contrast to the ALS classification case, the selected $\alpha$ values indicate high global sparsity over group sparsity, with $\alpha = 0.83$, 0.67, and 0.68, for the WH, HBN, and Cam-CAN datasets, respectively. The model weights are distributed over many different tracts and dMRI tissue properties (see Fig 3d–3f and S3–S8 Figs). This demonstrates that SGL is not coerced to produce overly sparse results when a more accurate model requires a dense selection of features. Furthermore, inspecting the portions of bundles with larger coefficients in Fig 3g–3i reveals that SGL selects informative regions where diffusion properties are different between the age quintiles.

As with ALS classification, we also compared SGL performance with results obtained using the pure lasso, elastic net, bundle-mean lasso, and Lasso PCR. The pure lasso models achieved $R^2 = 0.47$, 0.54, and 0.70 for the WH, HBN, and Cam-CAN datasets respectively using the same cross-validation strategy used to assess the SGL. The elastic net models achieved $R^2 = 0.46$, 0.59, and 0.73 for the WH, HBN, and Cam-CAN datasets, respectively. The bundle-mean lasso models achieved $R^2 = 0.33$, 0.22, and 0.55, respectively. And the Lasso PCR models achieved $R^2 = 0.54$, 0.57, 0.74, respectively. The SGL models' performance improvement over lasso is more modest than in the ALS classification case, which accords with the selected values

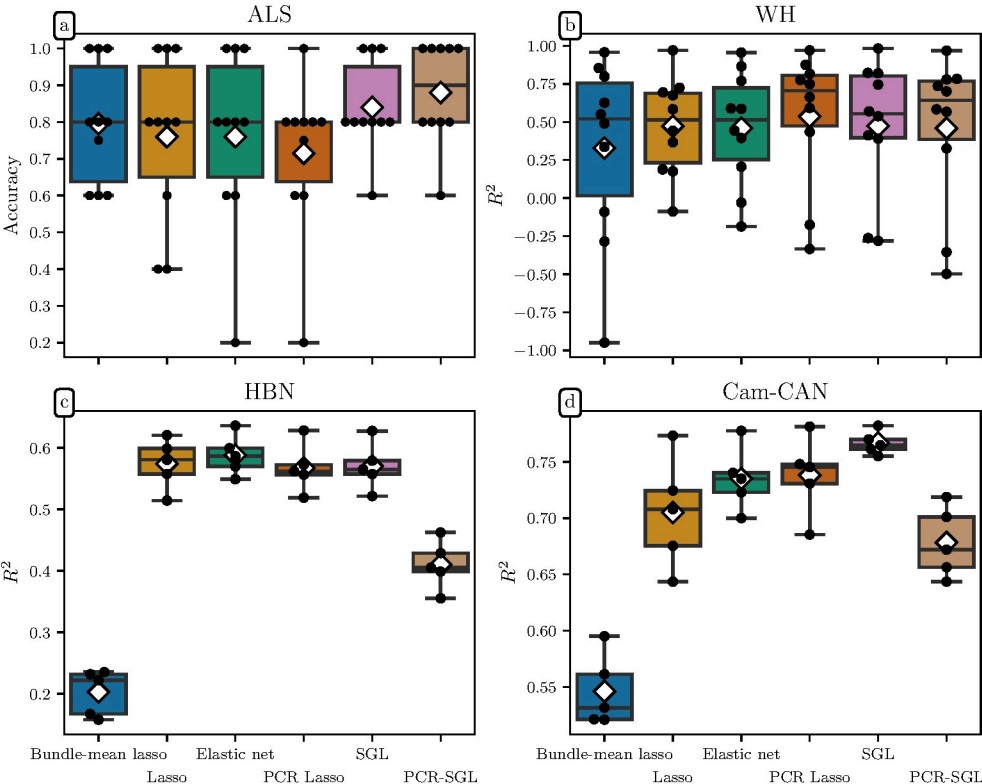

**Fig 4. Model performance across all datasets.** Each panel shows model performance measured on the test set for each cross-validation split, with each black dot representing a split, box plots representing the quartiles, and white diamonds representing the mean performance. The *y*-scale varies in each subplot. **(a)** Accuracy of test set predictions for the ALS dataset. Because group differences in ALS diagnosis are mostly confined to a single bundle, the group structure-preserving methods, SGL and PCR-SGL, outperform the other models. The remaining frames show coefficient of determination, $R^2$ in test sets for the **(b)** WH, **(c)** HBN, and **(d)** Cam-CAN datasets. Because aging affects the white matter globally, group structure-blind methods like elastic net and PCR Lasso perform well. Nonetheless, the SGL models show competitve predictive performance, adapting to a problem where group structure is not as informative. PCR-SGL performs poorly in this regime because its initial group-wise PC projection destroys between bundle covariance. The bundle-mean lasso performs poorly, demonstrating the value of along-tract profiling.

of the $\alpha$ hyperparameter. These SGL models were more lasso-like in their sparsity penalties so they are more lasso-like in their predictive performance.

Model performance across all four datasets and all six model types is summarized in Fig 4. In contrast to the ALS classification case, the PCR Lasso performs competitively in age regression, suggesting that aging is a significant source of global variance in the tract profiles. On the other hand, the PCR-SGL models destroy cross-bundle covariance information in their initial group-wise PC projection step (see Eq (4)), limiting performance for globally distributed phenomena. Conversely, the SGL models are able to adapt to this regime.

## Discussion

We present here a novel method for analysis of dMRI tract profile data that relies on the Sparse Group Lasso [46] to make accurate predictions of phenotypic properties of individual subjects while, simultaneously, identifying the features of the white matter that are most important for this prediction. The method is data-driven and it is broadly applicable to a wide range of research questions: it performs well in predicting both continuous variables, such as biological age, as well as categorical variables, such as whether a person is a patient or a healthy control.

In both of these cases, SGL out-performs previous algorithms that have been developed for these tasks [26, 44, 45]. The nested cross-validation approach used to fit the model and make both predictions and inferences from the model guards against overfitting and tunes the degree of sparseness required by the algorithm. This means that SGL can accurately describe phenomena that are locally confined to a particular anatomical location or diffusion property (e.g., FA in the CST) as well as phenomena that are widely distributed amongst brain regions and measured diffusion properties.

Specifically, we demonstrated that the algorithm correctly detects the fact that ALS, which is a disease of motor neurons, is localized to the cortico-spinal tract. This recapitulates the results of previous analysis of these same data, using a bundle-of-interest approach [26]. As with the original study, it is unclear whether our strong predictive performance is based on image properties that would be visible in patient scans or whether it is identifying a new sub-clinical disease biomarker. In contrast, for the analysis of biological age, the coefficients identified by the algorithm are very widely distributed across many parts of the white matter, mirroring previous results that show a large and continuous distribution of life-span changes in white matter properties [39]. We present age quintile bundle profiles and SGL $\beta$ coefficients for all four datasets in S3–S8 Figs. For age regression, substantial differences between the datasets preclude generalization across all three: (i) different acquisition parameters, which can be challenging to harmonize [47], (ii) different diffusion models, with DTI for the WH dataset and DKI for the HBN and Cam-CAN datasets, (iii) different age ranges and distributions (which is evident in the figure legends for Fig 3), with HBN being a developmental dataset, while WH and Cam-CAN are lifespan maturation datasets, and (iv) different anatomical extents, with the WH streamlines truncated to remain with the bundle's bounding regions of interest (the default behavior in the legacy mAFQ) and the HBN and Cam-CAN streamlines allowed to retain their full extent from whole-brain tractography (the default behavior in pyAFQ). In a biological characteristic that has widespread effects in the brain, validity is difficult to assess [48], so it is both unsurprising and vexing that we can do so well at predicting within each dataset, and yet do poorly in interpreting informative features across instruments.

We also implemented a union of the PCR Lasso and SGL approaches by first projecting each group onto its PC basis. This PCR-SGL approach transformed the feature space into a more parsimonious representation of its variance while also preserving group structure. But it was unable to efficiently represent cross-bundle covariance, which is an advantage of PCR Lasso. As a result, PCR-SGL performed well in ALS classification, where the white matter interactions are highly localized, and poorly in age regression, where the white matter interactions are global.

One drawback of our approach is also evident in the age regression results. There are portions of the bundle profiles in Fig 3g–3i for which the $\hat{\beta}$ coefficients are zero but for which the bundle profiles clearly contain an age differentiating signal. While SGL correctly identifies certain features as important, it is not guaranteed to identify *all* of the important features. It will identify a parsimonious set of important features, dense enough to predict the phenotype and sparse enough (at either the group or global level) to satisfy the sparsity constraints.

Another limitation of our approach is in its computational execution time, which, in our experience, is roughly five times slower than simpler models like the lasso. This slow-down is likely to be insignificant for "production" cycles in which researchers report their findings, but may be prohibitive for "development" cycles in which researchers repeatedly train models while adjusting their analysis pipeline. In practice, we circumvent this limitation by developing our analysis infrastructure with fast, simple models, and substituting slower, more performant models afterwards. The consistent scikit-learn-based application programming interface of AFQ-Insight provides a good basis for this pattern.

Taken together, our results demonstrate the promise of the group-regularized regression approach. Even at the scale of dozens of subjects, the results provided by SGL are both accurate, as well as interpretable [15]: tract profiling capitalizes on domain knowledge to engineer meaningful features; SGL scores these features based on their relative importance; enables a visualization of these feature importance scores in the anatomical coordinate frame of the bundles (e.g., Figs 2b and 3d–3f) and provides a means to understand model errors (e.g., Fig 2d and 2e). Thus, this multivariate analysis approach achieves high cross-validated accuracy for precision medicine applications of dMRI data and identifies relevant features of brain anatomy that can further our neuroscientific understanding of clinical disorders.

Neuroscience has entered an era in which consortium efforts are amassing large datasets of high-quality dMRI measurements to address a variety of scientific questions [40, 49–52], but the volume and complexity of these data pose substantial challenges. Tract profiling followed by analysis with SGL provides a promising approach to distill meaningful insights from the wealth of data measured in these efforts.

SGL has many other potential applications in neuroscience, because of the hierarchical and grouped nature of many data types that are collected in multiple sample points within anatomically-defined areas. For example, this method may be useful to understand the relationship between fMRI recordings and behavior, where activity in each voxel may co-vary with voxels within the same anatomical region and form features and groups of features. Similarly, large-scale multi-electrode recordings of neural activity in awake behaving animals are becoming increasingly feasible [53, 54] and these recordings can form features (neurons) and groups (neurons within an anatomical region).

The results we present here also motivate extensions of the method using more sophisticated cost functions. For example, the fused sparse group lasso (FSGL) [55] extends SGL to enforce additional spatial structure: smoothness in the variation of diffusion metrics along the bundles. As brain measurements include additional structure (for example, bilateral symmetry), future work could also incorporate overlapping group membership for each entry in the tract profiles [56]. For example, a measurement could come from the corpus callosum, but also from the right hemipshere. This would also require extending the cost function used here to incorporate these constraints. Similarly, unsupervised dimensionality reduction of tractometry data (e.g., [12]) could also benefit from constraints based on grouping, as our implementation of PCR-SGL suggests.

The method is packaged as open-source software called AFQ-Insight that is openly available at https://github.com/richford/AFQ-Insight, and provides a clear API to allow for extensions of the method. The sofware integrates within a broader automated fiber quantification software ecosystem: AFQ [5] and pyAFQ [57], which extract tract profile data from raw and processed dMRI datasets, as well as AFQ-Browser, which visualizes tract profiles data and facilitates sharing of the results of dMRI studies [58]. To facilitate reproducibility and ease use of the software, the results presented in this paper are also provided in https://github.com/richford/afq-insight-paper as a series of Jupyter notebooks [59].

## Materials and methods

### Data

Four different datasets were used here:

1. Diffusion MRI from a previous study of the corticospinal tract (CST) in patients with amyotrophic lateral sclerosis (ALS [26]), containing data from 24 ALS patients and 24 demographically matched healthy controls. These data were measured in a GE Discovery 750 3T

MRI scanner at the Institute of Bioimaging and Molecular Physiology in Catanzaro. Informed consent was provided as approved by the Ethical Committee of the University "Magna Graecia" of Catanzaro. Voxel resolution was $2 \times 2 \times 2 \text{mm}^3$ and 27 non-colinear directions were measured with a $b = 1000 \text{ s/mm}^2$. Data was preprocessed to correct for subject motion and for eddy currents. The diffusion tensor model [60] was fit in every voxel. We will refer to this dataset as ALS.

2. Diffusion MRI data from a previous study of properties of the white matter across the lifespan [39], containing dMRI data from 76 subjects with ages 6–50. These data were measured in a GE Discovery 750 3T MRI scanner at the Stanford Center for Cognitive and Neurobiological Imaging. The Stanford University IRB approved the procedures of this study. Informed consent was obtained from each adult participant, and assent for participation was provided by parents/guardians for children. Voxel resolution was $2 \times 2 \times 2 \text{mm}^3$ with 96 non-colinear directions measured with a $b = 2000 \text{ s/mm}^2$ and 30 non-colinear directions measured with a $b = 1000 \text{ s/mm}^2$. These data were acquired using a twice refocused spin echo sequence, in which there is sufficient time for eddy currents to subside between the application of the gradients and the image acquisition, so no eddy current correction was applied, but motion correction was applied before fitting the diffusion tensor model [60] in every voxel using a robust fit [61] on the $b = 1000 \text{ s/mm}^2$ shell only. We will refer to this dataset as WH.

3. Diffusion MRI data from the Healthy Brain Network pediatric mental health study [40], containing dMRI data from 1651 subjects with ages 5–21. These data were measured in 3T Siemens MRI scanners at two sites in the New York area: Rutgers University Brain Imaging Center and the CitiGroup Cornell Brain Imaging Center. Age distributions of participants in the different sites are in S9 Fig. Informed consent was obtained from each participant aged 18 or older. For participants younger than 18, written consent was obtained from their legal guardians and written assent was obtained from the participant. Voxel resolution was $1.8 \times 1.8 \times 1.8 \text{mm}^3$ with 64 non-colinear directions measured for each of $b = 1000 \text{s/mm}^2$ and $b = 2000 \text{s/mm}^2$. Preprocessing was performed using *QSIPrep* 0.12.1, which is based on *Nipype* 1.5.1 [62, 63], RRID:SCR_002502.

- *Anatomical data preprocessing* The T1-weighted (T1w) image was corrected for intensity non-uniformity (INU) using `N4BiasFieldCorrection` [64, ANTs 2.3.1], and used as T1w-reference throughout the workflow. The T1w-reference was then skull-stripped using `antsBrainExtraction.sh` (ANTs 2.3.1), using OASIS as target template. Spatial normalization to the ICBM 152 Nonlinear Asymmetrical template version 2009c [65], RRID:SCR_008796 was performed through nonlinear registration with `antsRegistration` [66], ANTs 2.3.1, RRID:SCR_004757, using brain-extracted versions of both T1w volume and template. Brain tissue segmentation of cerebrospinal fluid (CSF), white-matter (WM) and gray-matter (GM) was performed on the brain-extracted T1w using `FAST` [67], FSL 6.0.3:b862cdd5, RRID:SCR_002823.

- *Diffusion data preprocessing*
  Any images with a b-value less than 100 s/mm$^2$ were treated as a $b = 0$ image. MP-PCA denoising as implemented in MRtrix3's `dwidenoise` [68] was applied with a 5-voxel window. After MP-PCA, B1 field inhomogeneity was corrected using `dwibiascorrect` from MRtrix3 with the N4 algorithm [64]. After B1 bias correction, the mean intensity of the DWI series was adjusted so all the mean intensity of the b = 0 images matched across eachseparate DWI scanning sequence.
  FSL (version 6.0.3:b862cdd5)'s eddy was used for head motion correction and Eddy

current correction [69]. Eddy was configured with a $q$-space smoothing factor of 10, a total of 5 iterations, and 1000 voxels used to estimate hyperparameters. A linear first level model and a linear second level model were used to characterize Eddy current-related spatial distortion. $q$-space coordinates were forcefully assigned to shells. Field offset was attempted to be separated from subject movement. Shells were aligned post-eddy. Eddy's outlier replacement was run [70]. Data were grouped by slice, only including values from slices determined to contain at least 250 intracerebral voxels. Groups deviating by more than 4 standard deviations from the prediction had their data replaced with imputed values. Data was collected with reversed phase-encode blips, resulting in pairs of images with distortions going in opposite directions. Here, b = 0 reference images with reversed phase encoding directions were used along with an equal number of b = 0 images extracted from the DWI scans. From these pairs the susceptibility-induced off-resonance field was estimated using a method similar to that described in [71]. The fieldmaps were ultimately incorporated into the Eddy current and head motion correction interpolation. Final interpolation was performed using the `jac` method.

Several confounding time-series were calculated based on the *preprocessed DWI*: framewise displacement (FD) using the implementation in *Nipype* following the definitions by [72]. The DWI time-series were resampled to ACPC, generating a *preprocessed DWI run in ACPC space*.

- *MRtrix3 Reconstruction*
  Reconstruction was performed using *QSIprep* 0.12.1. Multi-tissue fiber response functions were estimated using the dhollander algorithm. FODs were estimated via constrained spherical deconvolution (CSD, [73, 74]) using an unsupervised multi-tissue method [75, 76]. Reconstruction was done using MRtrix3 [77]. FODs were intensity-normalized using mtnormalize [78].

  Many internal operations of *QSIPrep* use *Nilearn* 0.6.2 [79], RRID:SCR_001362 and *DIPY* [80]. For more details of the pipeline, see the section corresponding to workflows in *QSIPrep*'s documentation. We will refer to this dataset as HBN.

4. Diffusion MRI data from the Cambridge Centre for Ageing and Neuroscience (Cam-CAN) "CC700" dataset [41, 42], containing data from 640 subjects with ages 18–88. These data were measured on a 3T Siemens TIM Trio system and written informed consent was obtained from each participant. Voxel resolution was $2 \times 2 \times 2\text{mm}^3$ with 30 non-colinear directions measured for each of $b = 1000$ s/mm$^2$ and $b = 2000$ s/mm$^2$. The diffusion weighted images were acquired with a twice refocused spin-echo sequence and the same preprocessing and reconstruction pipelines used for the HBN dataset was applied to this data. We will refer to this dataset as Cam-CAN.

Data from the ALS and WH studies was processed in a similar manner, using the Matlab Automated Fiber Quantification toolbox (`mAFQ`, version 1.1 for WH and version 1.2 for ALS) [5]: streamlines representing fascicles of white matter tracts were generated using a determinstic tractography algorithm that follows the prinicpal diffusion direction of the diffusion tensor in each voxel (STT) [81]. Eighteen major tracts, which are enumerated in S1 Fig, were identified using multiple criteria: streamlines are selected as candidates for inclusion in a bundle of streamlines that represents a tract if they pass through known inclusion ROIs and do not pass through exclusion ROIs [82]. In addition, a probabilistic atlas is used to exclude streamlines which are unlikely to be part of a tract [83]. Each streamline is resampled to 100 nodes and the robust mean at each location is calculated by estimating the 3D covariance of the location of each node and excluding streamlines that are more than 5 standard deviations from the mean

location in any node. Finally, a bundle profile of tissue properties in each bundle was created by interpolating the value of MRI maps of these tissue properties to the location of the nodes of the resampled streamlines designated to each bundle. In each of 100 nodes, the values are summed across streamlines, weighting the contribution of each streamline by the inverse of the mahalanobis distance of the node from the average of that node across streamlines. This means that streamlines that are more representative of the tract contribute more to the bundle profile, relative to streamlines that are on the edge of the tract.

Data from the HBN and Cam-CAN studies were processed using the updated Python Automated Fiber Quantification toolbox (`pyAFQ`; [57]). In addition to demonstrating the our analysis pipeline is robust to changes in tractometry software, the use of the updated `pyAFQ` capitalized upon the following improvements over the legacy Matlab version: (i) the ability to ingest data provided in the BIDS format [84], and (ii) the calculation of diffusion kurtosis imaging (DKI [43]) metrics We will refer to the `mAFQ` and `pyAFQ` pipeline collectively as `AFQ`.

These processes create bundle profiles, in which diffusion measures are quantified and averaged along eighteen major fiber tracts, which are enumerated in S1 Fig. See S3 Fig of [57] for a depiction of these white matter bundles. Here, we use only the mean diffusivity (MD) and the fractional anisotropy (FA) of the diffusion tensor, but additional dMRI-based maps or maps based on other quantitative MRI measurements can also be used. The resulting feature space was the same for all four datasets, with the FA and MD metrics at each of 100 nodes in eighteen bundles comprising 3600 features per subject. These bundle profiles, along with the phenotypical data we wish to explain or predict, form the input to the SGL algorithm. In a domain-agnostic machine learning context, the phenotypical data comprise the target variables while the bundle profiles form the feature or predictor variables (see Fig 1e).

**Data harmonization for HBN.** For the multisite HBN study, we use the ComBat harmonization method to robustly adjust for site effects in the tract profiles. Initially designed to correct for site effects in gene expression studies [85], ComBat employs a parametric empirical Bayes approach to adjust for batch effects and has since been applied to multi-site cortical thickness measurements [86], multi-site DTI studies [87], and functional MRI data in the Adolescent Brain Cognitive Development Study (ABCD) [88]. We rely on the `neurocombat_sklearn` library [89], to apply ComBat in our scikit-learn analysis pipeline and present bundle profile site differences and ComBat correction in S10–S13 Figs.

## Sparse group lasso

Before fitting a model to the data, imputation and standardization are performed. Missing node values (e.g., in cases where `AFQ` designates a node as not-a-number) are imputed via linear interpolation. Care is taken not to interpolate across the boundaries between different bundles. Some diffusion metrics will have naturally larger variance than others and may therefore dominate the objective function and make the SGL estimator unable to learn from the lower variance metrics. For example, fractional anisotropy (FA) is bounded between zero and one and could be overwhelmed by an unscaled higher-variance metric like the mean diffusivity (MD). To prevent this we remove each feature's mean and scale it to unit variance (z-score) using the `StandardScaler` from scikit-learn [90]. The scaling parameters are learned only from the training data and then applied equally to the training and test data in order to prevent leakage of information between the testing and training sets [91].

After scaling and imputation, the tract profile data and target phenotypical data can be organized in a linear model:

$$y = \mathbf{X}\beta + \epsilon, \tag{1}$$

where $y$ is the phenotype—categorical, such as a clinical diagnosis, or continuous numerical, such as the subject's age. The tract profile data is represented in the feature matrix $\mathbf{X}$, with rows corresponding to different subjects, and columns corresponding to diffusion measures at different nodes within each bundle. The relationship between tractometric features and the phenotypic target is characterized by the coefficients in $\beta$. The error term, $\epsilon$ is an unobserved random variable that captures the error in the model. We denote our prediction of the target phenotype as $\hat{y}$ and the coefficients that produce this prediction as $\hat{\beta}$, so that

$$\hat{y} = \mathbf{X}\hat{\beta}, \tag{2}$$

without the error term, $\epsilon$. In general, the feature matrix $\mathbf{X}$ has $S$ rows and $B \times N \times M$ columns, where $S$ is the number of subjects, $B$ is the number of white matter bundles, $N$ is the number of nodes in each bundle, and $M$ is the number of diffusion metrics calculated at each node. Typically, $B = 18$, $N = 100$, and $2 \leq M \leq 8$, resulting in $\sim 4,000 - 14,400$ features. Conversely, many dMRI studies have between a few dozen and a few hundred subjects, yielding a feature matrix that is wide and short. Even in cases where more than a thousand subjects are measured (e.g., in the Human Connectome Project, where 1,200 subjects were measured [52]), the problem is ill-posed: the high dimensionality of this data requires regularization to avoid overfitting and generate interpretable results.

We propose that in addition to regularizing the coefficients in $\hat{\beta}$, we can also capitalize on our knowledge of the group structure of the bundle profile features in $\mathbf{X}$. The bundle-metric combinations form a natural grouping. For example, we expect that MD features within the left arcuate fasciculus will co-vary across individuals. Likewise for FA values within the right corticospinal tract (CST) and so on. This group structure is represented in Fig 1e, which depicts the linear model $\hat{y} = \mathbf{X}\hat{\beta}$. Thus, we seek a regularization approach that will fit a linear model with anatomically-grouped covariates, where we expect to observe both groupwise sparsity, where the number of groups (bundle/metric combinations) with at least one non-zero coefficients is small, as well as within-group sparsity, where the number of non-zero coefficients within each non-zero group is small. The sparse group lasso (SGL) is a penalized regression technique that satisfies these criteria [46]. It solves for a coefficient vector $\hat{\beta}$ that satisfies

$$\hat{\beta} = \min_{\beta} L_{\mathrm{mse}} + (1-\alpha)\lambda\sum_{\ell=1}^{G}\sqrt{p_\ell}\|\beta^{(\ell)}\|_2 + \alpha\lambda\|\beta\|_1,$$
$$\text{where} \quad L_{\mathrm{mse}} = \frac{1}{2}\left\|y - \sum_{\ell=1}^{G}\mathbf{X}^{(\ell)}\beta^{(\ell)}\right\|_2^2. \tag{3}$$

Here, $G$ is the number of groups, $\mathbf{X}^{(\ell)}$ is the submatrix of $\mathbf{X}$ corresponding to group $\ell$, $\beta^{(\ell)}$ is the coefficient vector for group $\ell$ and $p_\ell$ is the length of $\beta^{(\ell)}$. In the tractomtetry setting, $G = B \times M$ and $\forall\ell: p_\ell = 100$. The first term is the mean square error loss, $L_{\mathrm{mse}}$, as in the standard linear regression framework. The second and third terms encourage groupwise sparsity and overall sparsity, respectively. If $\alpha = 1$ the SGL reduces to the traditional lasso [92]. Conversely, if $\alpha = 0$ the SGL reduces to the group lasso [93]. Thus, the model hyperparameter $\alpha$ controls the combination of group-lasso and lasso. The hyperparameter $\lambda$ controls the strength of the regularization.

**SGL with principal components.** SGL may be combined with principal components regression (PCR-SGL) by performing dimensionality reduction separately for each group of covariates. Let

$$\mathbf{X}^{(\ell)} = \mathbf{U}\Sigma\mathbf{V}^T, \tag{4}$$

be the compact singular value decomposition (SVD) of $\mathbf{X}^{(\ell)}$, the $n \times p_\ell$ submatrix of $\mathbf{X}$ corresponding to group $\ell$. Here, $\Sigma$ is an $r \times r$ matrix, where $r = \min(n, p_\ell)$, that contains the non-zero singular values of $\mathbf{X}^{(\ell)}$. $\mathbf{V}^T$ is an $r \times p$ semi-orthogonal matrix containing the principal axes of $\mathbf{X}^{(\ell)}$. The product $\mathbf{Z} = \mathbf{U}\Sigma$ is an $n \times r$ matrix containing the principal component row vectors needed to reproduce $\mathbf{X}^{(\ell)}$ in the basis provided by $\mathbf{V}^T$.

Since this decomposition is performed separately for each group of covariates, the grouping information is preserved in the transformation from $\mathbf{X}$ to $\mathbf{Z}$. We may then build an SGL model relating $y$ and $\mathbf{Z}$,

$$\hat{y} \quad = \mathbf{Z}\hat{\theta}, \tag{5}$$

$$\hat{\theta} \quad = \min_{\theta} L_{\text{mse}} + (1 - \alpha)\lambda \sum_{\ell=1}^{G} \sqrt{p_\ell} \|\theta^{(\ell)}\|_2 + \alpha\lambda\|\theta\|_1, \tag{6}$$

$$\text{where} \quad L_{\text{mse}} \quad = \frac{1}{2} \left\| y - \sum_{\ell=1}^{G} \mathbf{Z}^{(\ell)}\theta^{(\ell)} \right\|_2^2. \tag{7}$$

The PCR-SGL coefficients $\hat{\theta}$ may be projected back onto the original feature space using $\hat{\beta} = \mathbf{V}\hat{\theta}$.

**Bagging meta-estimators.** The previous section describes a single SGL model. To further improve model performance, we create ensemble models composed of $m = 20$ individual SGL models using bootstrap aggregation (bagging) [94]. Bagging relies on the underlying assumption that some of the error in a single SGL model's prediction stems from a mismatch in the distributions of training data used to fit the model and test data used to evaluate its performance. To overcome this, bagging invokes the same base estimator (e.g. SGL) many times with different training sets, which are created by sampling the original training samples with replacement. The bagging meta-estimator's prediction is then the average of its constituent estimators' predictions. Likewise, when we report the hyperparameter values $\alpha$ and $\lambda$, or regression coefficients $\hat{\beta}$, we are refering to these values averaged over 20 estimators in the bagging meta-estimator.

**Incorporating target transformations.** Often, the target variable $y$ is not in the domain in which the linear model can be best fit to it. Eq (2) can be slightly modified as:

$$\hat{y} = f^{-1}(\mathbf{X}\hat{\beta}), \tag{8}$$

where the transformation function $f^{-1}$ characterizes the transform applied to the data before fitting the linear coefficients. This is similar to the use of a link function in a generalized linear model, but without the adoption of an exponential probability distribution [95]. For the WH, HBN, and Cam-CAN datasets, we use a logarithmic transform,

$$f(\hat{y}) = \ln(\hat{y}), \tag{9}$$

implemented using scikit-learn's `TransformedTargetRegressor` meta-estimator [90] with numpy's `log` and `exp` as the transform and inverse transform functions, respectively [96].

**Classification of categorical targets.** When the phenotypical target variable is categorical, as in the case of explaining or predicting the presence of a clinical diagnosis, the SGL is readily adapted to logistic regression, where the probability of a target variable belonging to an

arbitrary defined "true" class is the logistic function of the result of the linear model,

$$p(\hat{y} = 1) = \frac{1}{1 + \exp(\mathbf{X}\hat{\beta})}, \tag{10}$$

or equivalently, the mean squared error loss function in Eq (3) is replaced with the cross-entropy loss, which for binary classification is the negative log likelihood of the SGL classifier giving the "true" label:

$$L_{\mathrm{mse}} \rightarrow L_{\mathrm{log}} = -(y \log(p) + (1 - y) \log(1 - p)). \tag{11}$$

## SGL implementation, cross-validation and metaparameter optimization

Because the SGL is not specific to tractometry, we implemented its solution as a general-purpose Python package called `groupyr` [97]. Groupyr solves the cost function in Eq (3) using proximal gradient descent [98] by implementing a custom proximal operator and relying on the C-OPT optimization library [99], providing a fitted SGL model as a scikit-learn compatible estimator [100]. Groupyr also selects the hyperparameters $\alpha$ and $\lambda$ that yield the best cross-validated performance using either: (i) an exhaustive grid search of hyperparameter combinations, or (ii) sequential model based optimization using the scikit-optimize library [101].

To objectively evaluate model performance and guard against over-fitting, we used a nested cross-validation scheme, which is depicted for the binary classification case in Fig 5. The subjects (i.e. rows of the feature matrix $\mathbf{X}$ in Fig 1e and Eq (1)) are first shuffled and then decomposed into $k_0$ batches, hereafter referred to as folds. For the ALS and WH datasets, we used $k_0 = 10$ and for the HBN and Cam-CAN datasets, $k_0 = 5$. For each unique fold, we hold that fold out as the test$_0$ set and let the remaining data comprise the train$_0$ set, with the subscript indicating

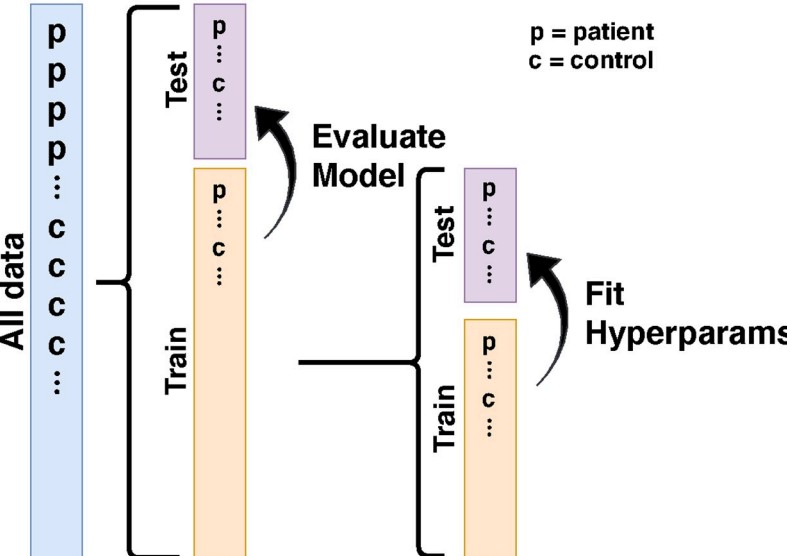

**Fig 5. Nested cross-validation.** We evaluate model quality using a nested $k$-fold cross validation scheme. At level-0, the input data is decomposed into $k_0$ shuffled groups and optimal hyperparameters are found for the level-0 training set. To avoid overfitting, the optimal hyperparameters are themselves evaluated using a cross-validation scheme taking place at level-1 of the decomposition, where each level-0 training set is further decomposed into $k_1 = 3$ shuffled groups. In the classification case, the training and test splits are stratified by diagnosis. For the ALS and WH data, $k_0 = 10$, while for the HBN and Cam-CAN data, $k_0 = 5$.

the depth of the nested decomposition. We further decompose each $\text{train}_0$ set into three folds, and again for each unique fold, we hold out that fold as the $\text{test}_1$ set and let the remaining data comprise the $\text{train}_1$ set. At level-1 of the decomposition, we fit an SGL model using fixed regularization meta-parameters $\alpha$ and $\lambda$, training the model using $\text{train}_1$ and evaluating the fit on $\text{test}_1$. We find the optimal values for $\alpha$ and $\lambda$ using sequential model-based optimization, implemented using the scikit-optimize `BayesSearchCV` class [101]. For continuous numerical $y$, `BayesSearchCV` searches for meta-parameter values that maximize the $R^2$ averaged over test sets. For binary categorical $y$ `BayesSearchCV` seeks to maximize the classification accuracy.

Using hyperoptimization, we find optimal regularization parameters and $\hat{\beta}$ for each $\text{train}_0$ set and then use those to predict values for data in $\text{test}_0$. Thus each subject in the dataset has a predicted phenotype derived from a model that never saw its particular subject's data. In the classification case, the shuffling into folds is stratified such that each fold has a population that preserves the percentage of each class found in the larger input data.

For each dataset, we also perform a randomization test by training similar models on copies of the data for which the rows of the target variable $y$ have been shuffled while the feature matrix $\mathbf{X}$ remains the same. This effectively destroys any relationship between the diffusion data and the outcome. Indeed, all of our models perform no better than random guessing. One should expect this since any better performance might indicate data leakage between train and test sets [91] or other common machine learning pitfalls.

## Software implementation

The full software implementation of the SGL approach presented here is available in a Python software package called `AFQ-Insight`, which is developed publicly in https://github.com/richford/afq-insight. The version of the code used to produce the results herein is also available in https://doi.org/10.5281/zenodo.4316000. AFQ-Insight reads the target and feature data that has been processed by AFQ from comma separated value (CSV) files conforming to the AFQ-Browser data format [58] and represents them internally as `DataFrame` objects from the pandas Python library [102]. The software provides different options for imputing missing data in the feature matrix. Missing interior nodes are imputed using linear interpolation. For missing exterior nodes, the user may choose between linear extrapolation and constant forward- or back-fill. Imputation uses only values from adjacent nodes in the same white matter bundle in the same subject so there is no danger of data leakage from other subjects. It uses the scikit-learn [90] library to decompose input data into separate test and train datasets, to scale each feature to have zero mean and unit variance, and to evaluate each fit in the hyperparameter search using appropriate classification and regression metrics such as accuracy, area under the receiver operating curve (AUC ROC), and coefficient of determination ($R^2$). Internally, `AFQ-Insight` uses the `groupyr` software library [97] mentioned above to solve the SGL.

## Supporting information

**S1 Fig. Bundle profiles and regression coefficients for the ALS dataset FA.** Diffusion metrics are plotted along the length of eighteen bundles: right corticospinal (CSTR), left corticospinal (CSTL), right uncinate (UNCR), left uncinate (UNCL), left inferior fronto-occipital fasciculus (IFOL), right inferior fronto-occipital fasciculus (IFOR), right arcuate (ARCR), left arcuate (ARCL), right thalamic radiation (ATRR), left thalamic radiation (ATRL), right cingulum cingulate (CGCR), left cingulum cingulate (CGCL), callosum forceps posterior (CFP), callosum forceps anterior (CFA), right inferior longitudinal fasciculus (ILFR), left inferior longitudinal fasciculus (ILFL), right superior longitudinal fasciculus (SLFR), and left superior

longitudinal fasciculus (SLFL). FA is plotted on the left $y$-axis while the $\hat{\beta}$ coefficients are displayed on the twin axis on the right-hand-side. SGL selected the right corticospinal tract (CSTR) as important and regularized coefficients in the CSTL. Yet, there are also group FA differences in the CSTL. This highlighted a potential drawback of the SGL method, discussed in the main text in the context of age regression. Namely, SGL is not guaranteed to identify *all* important features. In this case, if the diagnostic signal in the CSTL is redundant to that in the CSTR, SGL will regularize the CSTL features, thereby reducing its sparsity penalty without any corresponding increase in loss. This parsimony cuts both ways; it is a feature of the method when one seeks an efficient predictive model, but is a disadvantage of the method when one wants an exhaustive explanation of feature importance. We use the phrase "parsimony pitfall" to refer to the case when SGL regularizes away redundant but obviously important features. (TIF)

**S2 Fig. Bundle profiles and regression coefficients for the ALS dataset MD.** The scale of the $\hat{\beta}$-axis is identical to that used in S1 Fig, to facilitate the comparison of the relative importance of each metric.
(TIF)

**S3 Fig. Bundle profiles and regression coefficients for the WH dataset FA.** In contrast to the ALS classification case, the $\hat{\beta}$ coefficients are distributed widely through the brain, supporting the interpretation that aging is a large and continuous whole-brain process.
(TIF)

**S4 Fig. Bundle profiles and regression coefficients for the WH dataset MD.** This and S3 Fig demonstrate that SGL behaves much more like the lasso than the group lasso, as discussed in the main text. The parsimony pitfall is most evident in the IFOL and IFOR bundles.
(TIF)

**S5 Fig. FA bundle profiles and $\hat{\beta}$ coefficients for age regression in the HBN dataset.** Like the WH dataset, the $\hat{\beta}$ coefficients are distributed widely through the brain and SGL behaves more like the lasso than the group lasso.
(TIF)

**S6 Fig. MD bundle profiles and $\hat{\beta}$ coefficients for age regression in the HBN dataset.** In contrast to the WH results, the bundle profiles show different behaviors. For example the SLFL and SLFR bundle profiles in have different concavity. This is unsurprising, however, given the differences between these datasets (see also Discussion). The parsimony pitfall is most evident in the UNCL, UNCR, ARCL, SLFL, and SLFR bundles.
(TIF)

**S7 Fig. FA bundle profiles and $\hat{\beta}$ coefficients for age regression in the Cam-CAN dataset.** The $\hat{\beta}$ coefficients are distributed widely through the brain and SGL behaves more like the lasso than the group lasso. As before, one must be cautious about comparing bundle profiles and $\hat{\beta}$ coefficients between models. While the HBN and Cam-CAN datasets share the same diffusion model and refrain from clipping streamlines, the age distributions for the two are roughly disjoint, with the WH age distribution straddling the two.
(TIF)

**S8 Fig. Mean diffusivity (MD) bundle profiles and $\hat{\beta}$ coefficients for age regression in the Cam-CAN dataset.**
(TIF)

**S9 Fig. Age distributions are similar between the different HBN sites.** Rutgers Rutgers University Brain Imaging Center (RU) and the CitiGroup Cornell Brain Imaging Center (CBIC). (TIF)

**S10 Fig. FA bundle profiles exhibit strong site differences in the HBN dataset.** (TIF)

**S11 Fig. MD bundle profiles exhibit strong site differences in the HBN dataset.** (TIF)

**S12 Fig. Site differences in FA are eliminated by ComBat harmonization.** (TIF)

**S13 Fig. Site differences in MD are eliminated by ComBat harmonization.** (TIF)

## Acknowledgments

We would like to thank Scott Murray for a useful discussion of the SGL method and Mareike Grotheer and John Kruper for helpful comments on the manuscript. Data collection and sharing for this project was provided by the Cambridge Centre for Ageing and Neuroscience (CamCAN). CamCAN funding was provided by the UK Biotechnology and Biological Sciences Research Council (grant number BB/H008217/1), together with support from the UK Medical Research Council and University of Cambridge, UK. This manuscript was prepared using a limited access dataset obtained from the Child Mind Institute Biobank, The Healthy Brain Network dataset. This manuscript reflects the views of the authors and does not necessarily reflect the opinions or views of the Child Mind Institute.

## Author Contributions

**Conceptualization:** Adam Richie-Halford, Jason D. Yeatman, Noah Simon, Ariel Rokem.

**Data curation:** Adam Richie-Halford, Jason D. Yeatman.

**Formal analysis:** Adam Richie-Halford, Noah Simon.

**Funding acquisition:** Ariel Rokem.

**Investigation:** Adam Richie-Halford.

**Methodology:** Adam Richie-Halford, Noah Simon.

**Project administration:** Adam Richie-Halford, Ariel Rokem.

**Resources:** Ariel Rokem.

**Software:** Adam Richie-Halford.

**Supervision:** Jason D. Yeatman, Noah Simon, Ariel Rokem.

**Visualization:** Adam Richie-Halford, Jason D. Yeatman, Noah Simon, Ariel Rokem.

**Writing – original draft:** Adam Richie-Halford, Ariel Rokem.

**Writing – review & editing:** Adam Richie-Halford, Jason D. Yeatman, Noah Simon, Ariel Rokem.

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
