## [Decision Letter · Decision Letter 0]

26 Mar 2021

Dear Richie-Halford,

Thank you very much for submitting your manuscript "Multidimensional analysis and detection of informative features in human brain white matter" for consideration at PLOS Computational Biology. Three reviewers have provided their comments and they are all positive about the quality of your contribution. They ask, however, for some additional precision in the discussion of alternative methods and the description of the results that I encourage you to take into consideration, as they should help increase the impact of your work. We will be looking forward to receiving a revised version of your manuscript, taking these comments into account.

We cannot make any decision about publication until we have seen the revised manuscript and your response to the reviewers' comments. Your revised manuscript is also likely to be sent to reviewers for further evaluation.

Sincerely,

Roberto Toro

Associate Editor

PLOS Computational Biology

Daniele Marinazzo

Deputy Editor

PLOS Computational Biology

Dear Authors,

Thank you for submitting your article. Three reviewers have provided their comments and they are all positive about the quality of your contribution. They ask, however, for some additional precision in the discussion of alternative methods and the description of the results that I encourage you to take into consideration, as they should help increase the impact of your work. We will be looking forward to receiving a revised version of your manuscript, taking these comments into account.

Best regards,

Roberto Toro

Reviewer's Responses to Questions

**Comments to the Authors:**

Reviewer #1: This article describes a systematic evaluation of sparse group lasso (SGL) applied to tractometry output of dMRI-derived streamlines as a classification and individual differences method. Using SGL+tractometry in the classification setting, the authors reliably distinguish between ALS patients and matched controls at a classification accuracy of 83%, with the major contributing pathways to classification being the descending corticospinal (CST) pathways, as would be expected for an upper motor neuron disease. In the regression context, the authors found that they could reliably predict chronological age in 3 different data sets, with r^2’s ranging from 0.52-0.77. The authors conclude that this approach provides an effective way of identifying white matter phenotypes from dMRI data that also identifies contributing pathways relevant to model accuracy.

Overall, I found the results to be quite compelling. There are some outstanding questions about how this method relates to prior approaches and details about the modeling work itself that should be addressed. But in general I think this work will be a valuable contribution to the field.

MAJOR COMMENTS

- The authors make a compelling case that standard voxelwise approaches to developing brain-based phenotypes, particularly from dMRI data, are quite limited due to their high dimensionality and lack of specificity. One thing that is not clear up front, and could help in the interpretation, is that the authors are specifically talking about it in the context of decoding models (e.g., Y = group/individual measure), unlike many previous models (e.g., TBSS, connectometry) that are encoding models (e.g., Y= brain measure). For example, when I first read the Introduction, the discussion of prior approaches not characterizing specificity of bundles relevant to individual differences seemed incorrect as TBSS, connectometry, and even the initial tractometry approaches do this to some degree (and even control for the multiple comparisons). But the authors are correct that there are not many approaches that try to answer decoding questions while maintaining inferential specificity of individual pathways. Since the field is largely dominated by encoding models, I recommend making this clearer up front to avoid similar confusion in the future.

- Along the same lines as my prior comment, I wonder if the authors could compare their approach to the commonly used sparsity-constrained PCR (i.e., LASSO PCR) approach. This is becoming widely used in neuroimaging and has been applied specifically to dMRI data (see: 2) Powell, M. A., Garcia, J. O., Yeh, F. C., Vettel, J. M., & Verstynen, T. (2018). Local connectome phenotypes predict social, health, and cognitive factors. Network neuroscience, 2(1), 86-105. 2) Rasero, Javier, Amy Isabella Sentis, Fang-Cheng Yeh, and Timothy Verstynen. "Integrating across neuroimaging modalities boosts prediction accuracy of cognitive ability." PLOS Computational Biology 17, no. 3 (2021): e1008347.)*. I do not mean to vet directly against the SGL+tractometry method itself, but maybe add a section in the Discussion describing how this method compares to other common multivariate approaches used in computational neuroimaging like LASSO-PCR.

*Full disclosure: I am the senior author of these papers and am fully aware of how annoying it is for reviewers to ask that you cite their work. I’m not asking for citations per se, but a comparison of these PCR-based methods.

- The comparison of the SGL approach to traditional lasso seems limited. Traditional lasso will pick the best predictor out of a correlated set of predictors. Adding in the group constraint allows for keeping some degree of correlations/clustering in the final model (a good assumption when building predictors from brain imaging data sets). It seems to me that the appropriate comparison would be against a ridge model, since ridge allows for clustering of predictors, but in a different way than group lasso does. If I were to place money, my guess is that the performance of a ridge or elastic net model would sit somewhere in the middle of the lasso results and SGL results.

- There is substantial variability in the contributing pathways for age prediction across data sets. This seems incredibly important, but overlooked. One of the interpretive validations of the classifier results is that the pathways contributing to the classification make sense given the underlying pathologies of ALS. You’d assume that if age impacted the same white matter pathways, a validity test would be to show (i.e., quantify) that the same pathways predict age across data sets. But that’s not what we really see here beyond just a qualitative comparison of the arcuate fasciculus. Why is that? How similar or dissimilar are the tractometry results across datasets for this analysis?

- In the Methods it appears that the 3 data sets used in the age prediction differ substantially in their preprocessing routines. Is this a correct reading of the Methods? If so, how much could this variability explain the differences in model accuracies across data sets? Is there a way of understanding how specific preprocessing steps may impact model accuracy using this data?

MINOR POINTS

- Fig 1. Define acronyms (e.g., CFA, CST) in the caption.

- Since the arcuate is a major predicting pathway in the age prediction, it makes sense to show/highlight it in Fig. 1 for contrastive comparison, in the same way that CST pathways are highlighted in Fig. 2.

- Fig. 2b is not mentioned in the Results.

- Is one of the reasons that the prediction is so good in the ALS case is that there are lesions in the descending CST pathways? These should be visible in the B0 images (or T2 FLAIRs if they are available). If so, this should be added to the text as interpreting these results will be different if it is picking up on subclinical pathologies in the white matter signal or if it is just detecting visible lesions.

Reviewer #2: The paper presents a method (Sparse Group Lasso, SGL) to analyse along-tract data derived from diffusion MRI. Along-tract profiling of various microstructural measures typically poses statistical problems due to the high dimensionality nature of the data. Here, data from 4 databases were employed to test the proposed SGL framework under two scenarios: classification and regression. In particular, the authors looked at the ROC AUC scores to classify ALS subjects based on their tract profiles. The authors found that SGL generally improved classification over a previously published approach based on Random Forest, while detecting the CST – a WM bundle typically affected by ALS - as an important feature. For regression, the authors used SGL to predict brain age on 3 different datasets.

The paper is well written and well-structured. As a recommendation, I believe that the reader would benefit from a summary of the input format for each dataset (i.e., it is unclear if the feature space was the same across datasets regarding the number of tracts and metrics.)

My major comment concerns the classification scenario. Although results may seem impressive, I wonder how difficult was the task at first glance. The authors failed to provide a comparison with standard classification approaches relying on more standard tract-averaging approaches. Would a simple tract average of FA disentangle both groups (e.g., using a standard deviation threshold)? I appreciate that the framework identified the CST as an important feature in an unsupervised fashion, but in this “straightforward” clinico-radiological experiment, one expects the CST to be the driving feature for ALS. In other words, it is unclear if along tract-profiling provides added benefit for classification here (i.e., are 100 features per tract, per metric really necessary over 1 single feature per tract per metric?)

Minor comments:

“directional diffusion of water in each voxel [1].”

Stejskal, E. O., & Tanner, J. E. (1965). Spin diffusion measurements: spin echoes in the presence of a time‐dependent field gradient. The journal of chemical physics, 42(1), 288-292.

This approach is exhaustive, but statistical power is compromised by a multiple comparison problem [7, 9, 10].

Chamberland, M., Raven, E. P., Genc, S., Duffy, K., Descoteaux, M., Parker, G. D., ... & Jones, D. K. (2019). Dimensionality reduction of diffusion MRI measures for improved tractometry of the human brain. NeuroImage, 200, 89-100.

Fig. 1b) The red ROIs appear to be arbitrarily hand drawn over the corticospinal tract, which makes it confusing; e.g., were those starting-ending ROIs used to clip the CST profile or were they used for dissection only? I would say it is probably best to remove the drawing. The tract profile depicted in c) represents the traditional “bottom-to-top” CST FA profile with the well-known dip due to crossing fibers in the centrum semi-ovale. I would add the position along the tract along the x axis (c) so that uninitiated readers better appreciate the mapping from b to c.

Material, dataset 2: “…fitting the diffusion tensor model [51] in every voxel using a robust fit [52].” Please specify that this was done on the b=1000 s/mm2 shell only.

Typo: mahalnobis

Tractography: Please add details of missing parameters (step size, angular threshold, number of seeds, seeding method, number of streamlines per bundle per subject…)

Tractometry vs along tract profiling: I recommend paying special attention as to how both terms are mutually employed. In essence., Tractometry can refer to the quantitative mapping of multiples measures averaged over a set of tracts, as defined by Bells et al. 2011 and de Santis et al. 2014, whereas along-tract profiling specifically refers to the process of profiling the various MR-derived metrics, as defined by co-author J. Yeatman.

Overall, the framework provides a great addition to the existing AFQ software suite, and I find it worthy of publication at PLOS Computional Biology, conditional to the aforementioned comments.

Reviewer #3: The authors present a multidimensional analysis method of white matter diffusion-based measures. It is based on sparse group lasso and allows to obtain informative features. It has several advantages, such as being able to perform analysis along the tracts, to automatically determine relevant features, and to perform classification or regression.

The method is interesting and very sound and can be very useful for neuroscientists.

The manuscript is well written and explains quite clearly the method, experiments, and results.

However, I think that more detail could be given on the results obtained. These should be better explained and analyzed. The figures contain considerable information but are not sufficiently explained in the text.

In addition, for example, I do not see displays of the tracts, where the different zones detected as relevant are highlighted (especially for the study of ALS). There are some small plots of the centroids, but these are very small and do not show the actual tracts. What is CFA?

Also, there is not enough comparison with previous methods.

On the other hand, the limitations of the method with respect to previous methods need to be better discussed, considering different aspects such as complexity, processing time, the minimum sample size required, selection of hyperparameters. Is it always better to use this method?

Also, how the transformation function is specified?

Finally, there is not enough reference to the supplementary material in the text (or I did not see it). A discussion of the most important results could be in the main manuscript

**Have all data underlying the figures and results presented in the manuscript been provided?**

Reviewer #1: Yes

Reviewer #2: Yes

Reviewer #3: Yes

PLOS authors have the option to publish the peer review history of their article (what does this mean?). If published, this will include your full peer review and any attached files.

Reviewer #1: **Yes: **Timothy Verstynen

Reviewer #2: No

Reviewer #3: No
---

## [Decision Letter · Decision Letter 1]

31 May 2021

Dear Dr. Rokem,

We are pleased to inform you that your manuscript 'Multidimensional analysis and detection of informative features in human brain white matter' has been provisionally accepted for publication in PLOS Computational Biology.

Best regards,

Roberto Toro

Associate Editor

PLOS Computational Biology

Daniele Marinazzo

Deputy Editor

PLOS Computational Biology

Reviewer's Responses to Questions

**Comments to the Authors:**

Reviewer #1: The authors have done an exceptional job addressing my concerns.

Reviewer #2: The authors have addressed most of my concerns. I recommend acceptance of the manuscript, although relevant information regarding the tract-generation process is still missing. See original comment:

Tractography: Please add a simple description of the tractography parameters that were employed (step size, angular threshold, number of seeds, seeding placement (WM, GM...), number of streamlines per bundle per subject…).

Latex link seem to be broken + typo: (see Figs 3d to 3f and supporing figures in ??).

Reviewer #3: The authors addressed all the comments and have largely improved the manuscript.

I think it is a very good work and a valuable contribution to the field.

My only observation is about the supplementary material. It has too many separate files. I suggest creating a single supplementary file with all figures with captions, and comments.

**Have the authors made all data and (if applicable) computational code underlying the findings in their manuscript fully available?**

Reviewer #1: Yes

Reviewer #2: Yes

Reviewer #3: Yes

PLOS authors have the option to publish the peer review history of their article (what does this mean?). If published, this will include your full peer review and any attached files.

Reviewer #1: **Yes: **Timothy Verstynen

Reviewer #2: **Yes: **Maxime Chamberland

Reviewer #3: No

---

## [Editor Report · Acceptance letter]

24 Jun 2021

PCOMPBIOL-D-21-00231R1 

Multidimensional analysis and detection of informative features in human brain white matter

Dear Dr Richie-Halford,

I am pleased to inform you that your manuscript has been formally accepted for publication in PLOS Computational Biology. Your manuscript is now with our production department and you will be notified of the publication date in due course.

With kind regards,

Zsofi Zombor
